# Workplace discrimination as risk factor for long-term sickness absence: Longitudinal analyses of onset and changes in workplace adversity

Alice Clark[1☯]*, Sari Stenholm[2,3☯]*, Jaana Pentti[2,3,4], Paula Salo[5,6], Theis Lange[7], Eszter Török[1], Tianwei Xu[1,8], Jesper Fabricius[1,9], Tuula Oksanen[6,10], Mika Kivimäki[4,6,11], Jussi Vahtera[2,3‡], Naja Hulvej Rod[1‡]

1 Department of Public Health, Section of Epidemiology, University of Copenhagen, Copenhagen, Denmark, 2 Department of Public Health, University of Turku and Turku University Hospital, Turku, Finland, 3 Centre for Population Health Research, University of Turku and Turku University Hospital, Turku, Finland, 4 Faculty of Medicine, Clinicum, University of Helsinki, Helsinki, Finland, 5 Department of Psychology and Speech-Language Pathology, University of Turku, Turku, Finland, 6 Finnish Institute of Occupational Health, Helsinki, Finland, 7 Department of Public Health, Section of Biostatistics, University of Copenhagen, Copenhagen, Denmark, 8 Stress Research Institute, Stockholm University, Stockholm, Sweden, 9 Hammel Neurorehabilitation Centre and University Research Clinic, Hammel, Denmark, 10 Institute of Public Health and Clinical Nutrition, University of Eastern Finland, Kuopio, Finland, 11 Department of Epidemiology and Public Health, University College London, London, United Kingdom

☯ These authors contributed equally to this work.
‡ These authors are joint senior authors on this work.
* alcl@sund.ku.dk (AC); sari.stenholm@utu.fi (SS)

**Data Availability Statement:** The outcome of this study, long-term sickness absence, is obtained from the Finnish Social Insurance Institution's

## Abstract

Workplace discrimination may affect the health of the exposed employees, but it is not known whether workplace discrimination is also associated with an increased risk of long-term sickness absence. The aim of this study was to examine the longitudinal associations of changes in and onset of workplace discrimination with the risk of long-term sickness absence. Data on workplace discrimination were obtained from 29,597 employees participating in survey waves 2004, 2006, 2008 and/or 2010 of the Finnish Public Sector Study. Four-year changes in long-term sickness absence (≥10 days of medically certified absence with a mental or non-mental diagnosis) were assessed. This covered successive study waves in analyses of onset of workplace discrimination as well as fixed effect analyses of change in workplace discrimination (concurrent i.e. during the exposure year and 1-year lagged i.e. within one year following exposure), by using each employee as his/her own control. The risk of long-term sickness absence due to mental disorders was greater for employees with vs. without onset of workplace discrimination throughout the 4-year period, reaching a peak at the year when the onset of discrimination was reported (adjusted risk ratio 2.13; 95% confidence interval (CI) 1.80–2.52). The fixed effects analyses showed that workplace discrimination was associated with higher odds of concurrent, but not 1-year lagged, long-term sickness absence due to mental disorders (adjusted odds ratio 1.61; 95% CI 1.33–1.96 and adjusted odds ratio 1.02; 95% CI 0.83–1.25, respectively). Long-term sickness absence due to non-mental conditions was not associated with workplace

register and linked to survey data by using the unique personal identification number. According to the Finnish law, we are not allowed to share identified sensitive health data to other researchers. We are allowed to share anonymised questionnaire data of the Finnish Public Sector Study by application for with bona fide researchers with an established scientific record and bona fide organisations. For information about the Finnish Public Sector Study contact Prof. Mika Kivimaki mika.kivimaki[at]helsinki.fi / PI of the Finnish Public Sector study Dr. Jenni Ervasti jenni.ervasti[at]ttl.fi.

**Funding:** This work was supported by NordForsk, the Nordic Research Program on Health and Welfare [grant no. 75021 to NHR], Project on Psychosocial Work Environment and Healthy Ageing and the Danish Working Environment Foundation [grant no. 13-2015-09 to NHR]. MK was supported by grants from NordForsk, the UK Medical Research Council [grant no. K013351], the Academy of Finland [grant no. 311492], the Finnish Work Environment Foundation and a Helsinki Institute of Life Science (HILIFE) fellowship. The funders had no role in study design, data collection and analysis, decision to publish, or preparation of the manuscript.

**Competing interests:** The authors have declared that no competing interests exist.

discrimination. In conclusion, these findings suggest that workplace discrimination is associated with an elevated risk of long-term sickness absence due to mental disorders. Supporting an acute effect, the excess risk was confined to the year when workplace discrimination occurred.

## Introduction

A large body of literature has linked various adverse psychosocial work characteristics, including job strain, effort-reward imbalance, job insecurity, and long working hours, to increased risk of physical and mental chronic conditions [1–5]. These factors characterize unhealthy features of work tasks and organization of work. In addition, negative interpersonal relations at work have been hypothesized to predispose employees to ill health.

Workplace discrimination (WD) is a specific social stressor, being typically both unpredictable and uncontrollable. Among the general working population in Finland, 11% of women and 6% of men reported experiencing discrimination at their workplace [6]. As such, discrimination has been associated with mental and physical health complaints including psychological distress, depression and anxiety, cardiometabolic diseases [7–12] as well as lower job satisfaction, organizational commitment and work efficiency [12–16], thus yielding a great cost for the individual and society as a whole. Further, prior studies point to a potential link between workplace discrimination and the risk of sickness absence [12,13,17–22]. Sickness absence, especially when long-term, is considered as a measure employee health and well-being as well as a correlate of workplace productivity [23–26]. However, studies of workplace discrimination and sickness absence are few, and suffer from a number of methodological problems. Current evidence relies almost exclusively on cross-sectional data and self-reported information on sickness absence [12,13,18–22]. We are aware of no longitudinal studies on this topic.

The majority of studies on the importance of psychosocial work environment for employee health and well-being, including those on workplace discrimination, are based on exposure assessment at one, often arbitrary, time-point, and longitudinal analyses assessing the importance of the timing and duration of exposure remain scarce. The 'one measurement approach' fails to account for at least two common methodological concerns within psychosocial work environment research.

First, the important fact that employees who experience adverse psychosocial circumstances at work may quit their job as a means of coping [13]. Therefore, the effect of workplace adversities is likely to differ between employees remaining at the workplace and those who quit–possibly because of those adversities–leading to biased associations between WD and health.

Second, stable individual characteristics such as personality traits are important in determining how psychosocial exposures are experienced and reacted to by the individual employee, and research has shown that employees who report negative social relations at work generally differ from other employees in terms of psychological characteristics such as neuroticism, aggressiveness and negative affectivity [27–29]. With such traits also being predictive of employee health and well-being [30–33], failing to control for this may be an important source of bias. Given the intangible concept of person-stable characteristics, even studies adjusting for personality traits may not adequately address this potential source of bias.

To address these two methodological shortcomings, we combined longitudinal survey with register data from national registers from Finland and applied robust methodological approaches in assessing in the associations of onset of WD and changes in WD with risk of subsequent long-term sickness absence (LTSA). First, to reduce issues of selection bias, we

analyzed longitudinal data to compare changes in LTSA between employees with and without onset of workplace discrimination in a population in which all were free of that exposure at the first assessment. Second, we compared within-person changes in LTSA before and after a change in exposure to workplace discrimination to account for stable factors, such as personality, affecting the exposure to (confounding) and the reporting of (misclassification) workplace discrimination using each employee as his/her own control in fixed effects models.

## Materials and methods

### Study population

Participants were from the Ten town study, a part of the Finnish Public Sector Study (FPS) survey cohort, initiated in 1997. The survey cohort is nested within the FPS register cohort consisting of all Finnish public sector employees in ten towns and five hospital districts who had at least six months of employment between 1991 and 2005 [34,35]. The Ethics committee of Helsinki and Uusimaa Hospital District approved the study (registration number HUS/1210/2016). Participants were provided information about the study and response to the survey questionnaire was considered as a consent to participate. From 2004 the public sector employees working in the ten towns have been invited to biannual surveys including questions about work related factors.

Repeated information on workplace discrimination from waves 2004 (n = 32,197, response rate 65%), 2006 (n = 34,418, response rate 69%), 2008 (n = 38,838, response rate 70%) and 2010 (n = 37,651, response rate 69%) was used to assess the associations of onset of and changes in workplace discrimination with the risk of LTSA. To be eligible for the study, participants had to respond to at least two survey waves (successive waves were required for the assessment of onset) during which they were actively working for a minimum of 50% of the time and provide information on workplace discrimination in both surveys. In addition, linkage to the national health register on LTSA was required. These criteria were fulfilled by 32,519 employees. Since some participants had participated in all four study waves, they contributed observations to two separate two-wave data cycles (S1 Table). Given this, the 32,519 participants contributed to a total of 44,704 observations.

The selection of the study samples for the individual analyses of onset of and change in workplace discrimination is illustrated in S1 Fig.

For analyses of onset of workplace discrimination, 2811 observations with missing information on covariates (6% of the observations) were excluded. Among the 41,893 remaining observations, 2480 were excluded due to exposure to workplace discrimination at baseline (2004 and/or 2008) leaving 39,413 observations (in 29,597 employees) eligible for the analyses of onset of workplace discrimination. The structure of the data-cycles and follow-up is illustrated in S1 Table.

For analyses of change in workplace discrimination, 1125 observations with missing information on time-varying covariates (1% of the observations) were excluded. Due to the application of the fixed effect models, only observations with change in outcomes around the four waves where discrimination was measured were included in the analyses. The final number of employees and observations for analyses is illustrated in S1 Fig.

### Workplace discrimination

Perceived WD was measured with a single item: "I am exposed to discrimination at my workplace" with responses on a 5-point Likert scale: "Completely agree" (1), "Somewhat agree" (2), "Do not agree or disagree" (3), "Somewhat disagree" (4) and "Completely disagree" (5) [36]. Workplace discrimination was dichotomized as "yes" (response options 1–2) and "no"

(response options 3–5). In 2008 and 2010, the surveys included information regarding potential causes of workplace discrimination with the following categories: discrimination due to age, sex, ethnicity, education, opinion, and position.

### Long-term sickness absence

Using employees' unique personal identification numbers, we linked participants to the records of medically-certified LTSA from the Finnish Social Insurance Institution register, including dates and diagnoses for LTSA spells lasting 10 or days or more. This register covers the entire working population including public and private sector workers, but does not include sickness absence due to maternity leave and caring for a sick child. We distinguished between LTSA with an underlying mental (ICD-10 F01-F99) and non-mental diagnosis (other ICD-10 codes), and only primary diagnosis of LTSA was considered. In the text, terms mental LTSA and non-mental LTSA are used.

In the analyses with *onset of WD* as the exposure, the annual risk of LTSA (one or more episodes) for the groups with and without onset of WD was assessed. The risk of LTSA was calculated from the year pertaining to the first reporting of WD i.e. year 0 (where no employees were exposed) to year 3 (one year after the second study wave when WD was assessed). The study design is illustrated in the S1 Table.

In analyses with *change in WD* as the exposure, two different LTSA outcomes were defined: Concurrent LTSA was defined as sickness absence spells within the year pertaining to the reporting of exposure, and 1-year lagged LTSA was defined as sickness absence spells initiated during the year that followed the assessment of exposure. The study design is illustrated in the S1 Table.

### Covariates

Information on individual and work unit characteristics was drawn from surveys, employers' records, and national registers. Potential confounders were identified based on associated factors found in research literature and the methods of directed acyclic graphs [37] (See S2 Fig).

Individual level covariates included age, sex, chronic disease (diabetes, angina, acute myocardial infarction, chronic obstructive pulmonary disease, asthma, or cancer; no/yes), type of job contract (permanent/temporary), shift work (yes/no) and occupational grade (upper-grade non-manual, lower-grade non-manual, and manual workers) based on the ISCO-88 and the Occupational Title Classification of Statistics Finland. Information on alcohol consumption (abstainer, 1–16 units/week for women and 1–24 units/week for men, >16 units/week for women and >24 units/week for men) and body mass index (BMI) ($<25$, 25–30, $\geq 30$ kg/m$^2$) was obtained from the surveys. Psychological distress was assessed using the 12-item General Health Questionnaire ($<4$ vs $\geq 4$ symptoms) [38]. The six-item Trait Anxiety Inventory [39] was used to assess liability to anxiety and was analyzed on a continuous scale based on the average of responses 1–4.

Work unit level covariates included work unit size ($<20/\geq 20$ employees), work unit gender distribution (female dominated defined as <33% men, male dominated defined as <33% women, and balanced defined as $\geq 33$% women and men), and work unit proportion of temporary employees (above or below the median of 5%). Work unit level covariates were constructed by using information on the characteristics of the eligible population for each work unit.

### Statistical analysis

Two different analytic approaches were used to determine the relationship between 1) onset of WD and LTSA; and 2) changes in WD and LTSA, as outlined below.

**Analyses of onset of workplace discrimination.** The analyses were based on a four-year observation period, ranging from year 0 (where none was exposed to discrimination) to year 3 (one year after reported onset of WD). This time frame was chosen, because the FPS survey did not include question regarding the specific date of the onset of discrimination. It is therefore only known that the onset took place between two successive study waves. The annual average prevalence of LTSA and the corresponding 95% CIs were estimated for employees with and without onset of WD using a repeated-measures log-binomial regression and a generalized estimating equation (GEE) model to account for the intra-individual correlation between measurements and across data-cycles [40]. If this model did not converge, we used a Poisson distribution and a log link to assess the risk of sickness absence. The estimates were pooled across the two data-cycles, which contributed to an increased statistical efficiency of the analyses. Results are presented as risk ratios (RR) and 95% confidence intervals (CI).

The analyses were adjusted for individual and work-unit level confounders at baseline: age, sex, chronic disease, job contract, normal working hours, occupational grade, work unit size, gender distribution, proportion of temporary employees, alcohol consumption, BMI, psychological distress, and trait anxiety.

**Analyses of changes in workplace discrimination.** To control for person-stable characteristics (measured as well as unmeasured) a fixed-effect approach was used to assess the average within-person change in LTSA related to WD across the four waves.

Odds ratios (OR) of LTSA were assessed as the risk of having one or more episodes of LTSA occurring concurrently (within the exposure year) and with one-year lag (within one year after exposure) in conditional logistic regression models with the corresponding 95% CIs. Due to the within-subject comparisons, the covariates include only the potential time-varying individual and work-unit level confounders: age, job contract, work unit size, gender distribution, and proportion of temporary employees. Results are presented as ORs and 95% CI's.

**Analyses of cause-specific workplace discrimination.** Cause specific WD (age, sex, ethnicity, education, opinion, or position), was analyzed using data from the 2008 and 2010 study waves. The cause-specific onset and changes in WD and LTSA of mental and non-mental origin were examined separately as described earlier.

**Supplementary analyses.** To assess the robustness of findings, several supplementary analyses were carried out. First, previous LTSA may affect both subsequent LTSA and be a cause of discrimination. Therefore, the occurrence of concurrent LTSA was included as a covariate in analyses of one-year lagged LTSA in the fixed effect analyses of change in workplace discrimination, and as an additional cofounder in the analyses of onset.

As the time-varying health-related factors alcohol consumption, BMI and psychological distress may be either a cause or a consequence of discrimination, these were included as additional potential confounders in the fixed effects analyses available in the 2004 and 2008 study waves.

To explore the potential risk of reverse causation of the association between WD and LSTA, an additional analysis of the effect of LTSA on onset of WD was performed.

To examine the proposed source of bias from discriminated employees leaving their job, an additional analysis of the association between onset of discrimination and turnover was performed. Turnover was defined as working less than 50% of the time following onset of WD (year 3).

Finally, due to missing data on covariates, 6% of the observations for the onset analysis and 1% of the observations for the change analysis were excluded from the analysis. Sensitivity analysis for the main analyses were conducted by using mean imputation method in the full dataset in which missing covariates were replaced with their overall estimated mean. Statistical software SAS version 9.4 (SAS Institute Inc., Cary, NC, USA) was used for all the analyses. The level of statistical significance was set at 5%.

## Results

Baseline characteristics of the study population are presented in Table 1. Employees were between 18 and 66 years of age at baseline (mean 47 years). The vast majority were women (78%) reflecting the gender-distribution of the Finnish public sector employees. Among the 39,413 employee-observations from the group of 29,597 participants not exposed to WD at the baseline, 1,973 (5%) employee-observations among 1,920 participants indicated WD at the second time-point. The most common cause of WD were opinion, followed by education and position.

Relative to employees not being discriminated, those experiencing onset of WD were more likely to be male, have an intermediate occupational grade, work in shifts, have more chronic medical disorders, be obese and have more psychological distress at baseline, i.e. before onset of discrimination. The distribution of work unit size was comparable between those experiencing onset of discrimination and those not. Meanwhile, a higher proportion with onset of workplace discrimination worked at male dominated or gender balanced work units, and a lower proportion worked in work units with ≥5% temporary employees.

Table 1. Baseline characteristics of the Finnish Public Sector Study sample (in the onset analysis).

|  | All observations | Not discriminated | Onset of discrimination |
|---|---|---|---|
|  | (n = 39,413) | (n = 37,440) | (n = 1,973) |
| Causes of discrimination[a] |  |  |  |
| Age-discrimination | 552 (3) | - | 177 (25) |
| Sex-discrimination | 281 (1) | - | 99 (16) |
| Discrimination because of ethnicity | 41 (0.2) | - | 19 (3) |
| Discrimination because of education | 792 (4) | - | 222 (29) |
| Discrimination because of position | 674 (3) | - | 225 (30) |
| Discrimination because of opinion | 1081 (5) | - | 375 (41) |
| Employee characteristics |  |  |  |
| Women, n (%) | 30,870 (78) | 29,362 (78) | 1,508 (76) |
| Mean age (SD) | 47 (9) | 47 (9) | 48 (8) |
| Psychological distress, n (%) | 8,759 (22) | 8104 (22) | 655 (33) |
| Mean trait anxiety (SD) | 1.92 (0.54) | 1.91 (0.54) | 2.06 (0.62) |
| Chronic medical conditions[b], n (%) | 7,308 (19) | 6843 (18) | 465 (24) |
| Obese, n (%) | 5,936 (15) | 5,559 (15) | 377 (19) |
| High alcohol consumption, n (%) | 4,016 (10) | 3,816 (10) | 200 (10) |
| Occupational grade, n (%) |  |  |  |
| High | 15,299 (39) | 14,608 (39) | 691 (35) |
| Intermediate | 9,828 (25) | 9,273 (25) | 555 (28) |
| Low | 14,286 (36) | 13,559 (36) | 727 (37) |
| Temporary employment, n (%) | 2,685 (7) | 2534 (7) | 151 (8) |
| Shift-work, n (%) | 8,495 (22) | 8038 (21) | 457 (23) |
| Work-unit characteristics |  |  |  |
| ≥5% work unit temporary employment, n (%) | 20,184 (51) | 19,147 (51) | 1037 (53) |
| Work units with <20 employees, n (%) | 15,045 (38) | 14,315 (38) | 730 (37) |
| Gender distribution, n (%) |  |  |  |
| Female dominated; <33% men | 30,005 (76) | 28,640 (77) | 1365 (69) |
| Balanced; ≥33% women and men | 4,740 (12) | 4434 (12) | 306 (16) |
| Male dominated; <33% women | 4,668 (12) | 4366 (12) | 302 (15) |

[a] based on information from 2008 and 2010 where the information was available.

[b] ≥1 chronic disorders.

## Workplace discrimination and long-term sickness absence due to mental disorders

**Analyses of onset of workplace discrimination.** Fig 1 illustrates the prevalence of LTSA due to mental disorders for employees with and without onset of WD from year 0 to year 3. At year 0, where no one was exposed to WD, the prevalence of LTSA were comparable between the two exposure groups, with a prevalence of 2.7% and 2.3% for those with and without later onset of WD, respectively.

The prevalence of LTSA differed markedly between the groups during the 4-year follow-up (Fig 1). Among employees not experiencing WD, the prevalence of LTSA due to a mental disorders was almost constant over the 4-year period. Meanwhile, in employees with onset of WD there was an increase in LTSA due to mental disorders already at year 1. This difference in LTSA levels between the two exposure groups reached its peak at year 2, the time point when the onset of WD was reported (RR = 2.13; 95% CI 1.80–2.52).

**Analyses of changes in workplace discrimination.** As seen in Table 2, a short-term higher risk associated with WD was also observed in the fixed effect analyses. The odds of concurrent LTSA due to mental disorders were higher (OR = 1.61; 95% CI 1.33–1.96) in the year when WD was reported compared to the year(s) when WD was not reported. This finding did not persist in analyses of 1-year lagged LTSA (OR = 1.02; 95% CI 0.83–1.25).

## Workplace discrimination and long-term sickness absence due to non-mental conditions

**Analyses of onset of workplace discrimination.** As seen in Fig 1, the group with and without onset of WD differed slightly in the prevalence of non-mental LTSA at year 0

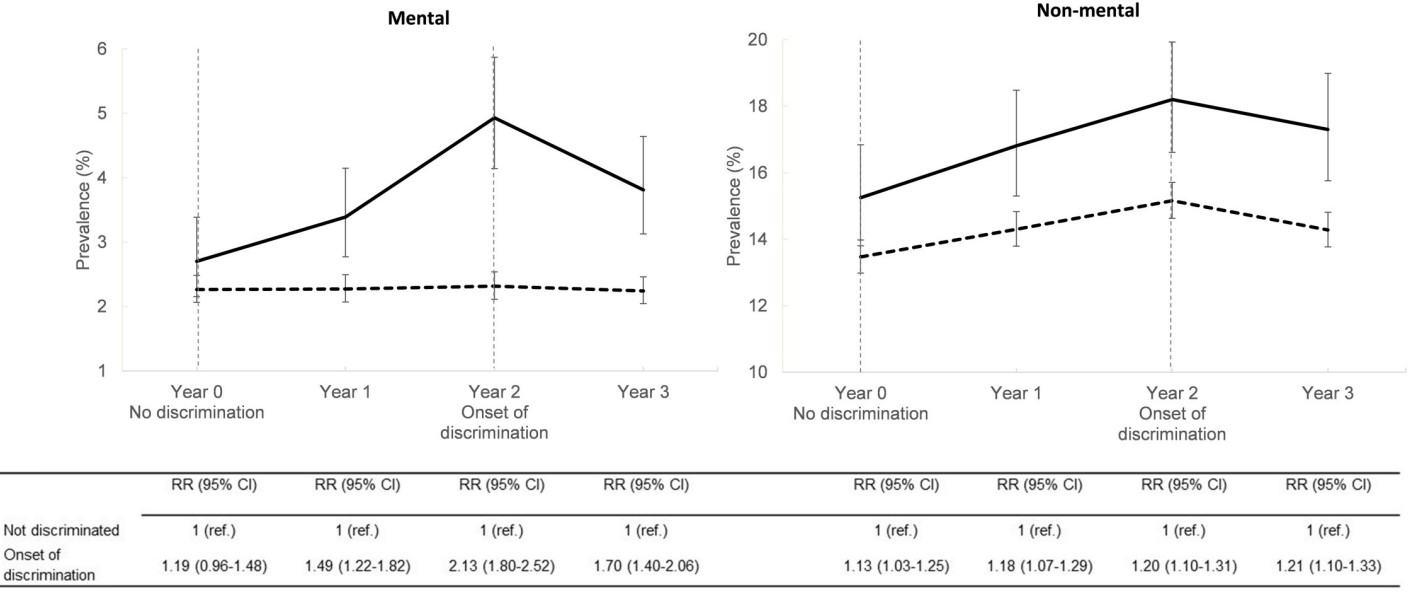

**Fig 1. Prevalence of long-term sickness absence among employees with and without onset of workplace discrimination as well as the risk ratio (RR) and 95% CI's associated with onset of workplace discrimination.** Log-binomial regression analyses adjusted for the following variables at year 0: Age, sex, chronic disease, psychological distress, anxiety, BMI, alcohol consumption, shift work, employment contract, occupational grade, work-unit size, work-unit temporary employment, work-unit gender distribution, as well as data-cycle number. Solid lines represents those with onset of workplace discrimination and dashed lines represents those who were not discriminated.

**Table 2. Changes in workplace discrimination for concurrent and 1-year lagged long-term sickness absence.**

| | | Concurrent sickness absence[a] | | 1-year lagged sickness absence[b] | |
|---|---|---|---|---|---|
| | Cause of sickness absence | Observations | OR [c] (95% CI) | Observations | OR [c] (95% CI) |
| Discrimination vs. no discrimination | Mental | 6821 | 1.61 (1.33–1.96) | 6621 | 1.02 (0.83–1.25) |
| Discrimination vs. No discrimination | Non-mental | 26,784 | 1.06 (0.95–1.19) | 26,906 | 1.11 (0.99–1.24) |

[a] One or more spells of absence during the exposure year.

[b] One or more spells within one year after exposure.

[c] Conditional logistic regression analyses adjusted for age, job contract, work-unit gender distribution, work-unit size, work-unit temporary employment.

(RR = 1.13; 95% CI 1.03–1.25), i.e. before onset of WD. This difference persisted throughout the follow-up as illustrated by the parallel lines for those with and without onset of WD, respectively.

**Analyses of changes in workplace discrimination.** The results showed comparable occurrence of non-mental LTSA when employees reported exposure to WD and when they did not, as seen in Table 2.

## Causes of workplace discrimination and long-term sickness absence

Analyses on the causes of WD showed a similar pattern as the main analyses. No difference in LTSA due to mental disorders were observed at year 0 among employees with and without later onset of WD for all cause, except for discrimination due to education and opinion, where the groups differed already at year 0 (Table 3). The exposure groups diverged at the following years with a higher occurrence of mental LTSA among those experiencing an onset of WD for all cause except sex-discrimination. Risk of LTSA due to mental disorders at year 3 was most pronounced for WD due to ethnicity (RR = 3.20; 95% CI 1.21–8.48).

Results of the fixed effect analyses supported these findings with a higher occurrence of concurrent, but not 1-yr lagged mental LTSA associated with all causes of discrimination apart from sex-discrimination. Noticeably, the occurrence of concurrent mental LTSA was tenfold associated with discrimination due to ethnicity, but with an extreme 95% CI due to the low prevalence of this exposure (OR = 10.56; 95% CI 1.36–82.14) (Table 4).

In terms of LTSA due to non-mental conditions, the risk of LTSA was consistently slightly higher among employees with onset of WD due to age, position and opinion compared to those without onset of WD (Table 3). Results pointed to a higher prevalence of LTSA following onset of WD due to ethnicity, with a twofold risk of LTSA at year 3 (RR = 2.12; 95% CI:1.43–3.13). The higher prevalence of LTSA associated with WD due to ethnicity, was also apparent in the fixed effect analysis in the year following exposure (1-year lagged OR = 3.09; 95% CI: 1.13–8.44) (Table 4). Similar to the main analyses, there was no association between WD for any other cause and LTSA due to non-mental conditions.

## Supplementary analyses

Additional adjustment for prior LTSA, or the potential time-varying confounders alcohol, BMI and psychological distress, did not change the findings of either the relations with onset of or change in WD, as seen in Tables 2 and 3.

Additional analyses of the potential reverse causation between LTSA and onset of WD supported a likely circular association with mental LTSA. Here LTSA due to mental disorders (but not due to non-mental condition) was associated with higher odds of subsequent exposure to WD (OR = 1.46; 95% CI 1.09–1.97) (S2 Table).

**Table 3. Onset of cause-specific workplace discrimination and annual risk of mental and non-mental diagnosed long-term sickness absence.**

| | | Mental diagnosed sickness absence | | | | Non-mental diagnosed sickness absence | | | |
|---|---|---|---|---|---|---|---|---|---|
| | | Year 0 | Year 1 | Year 2 | Year 3 | Year 0 | Year 1 | Year 2 | Year 3 |
| | Observations | RR [a] (95% CI) | RR a (95% CI) | RR a (95% CI) | RR a (95% CI) | RR a (95% CI) | RR a (95% CI) | RR a (95% CI) | RR a (95% CI) |
| Cause-specific workplace discrimination | | | | | | | | | |
| Age-discrimination | | | | | | | | | |
| Not discriminated | 18756 | 1 (ref.) | 1 (ref.) | 1 (ref.) | 1 (ref.) | 1 (ref.) | 1 (ref.) | 1 (ref.) | 1 (ref.) |
| Onset of discrimination | 552 | 1.18 (0.78–1.80) | 1.32 (0.87–2.00) | 1.65 (1.13–2.42) | 1.68 (1.12–2.51) | 1.19 (1.00–1.42) | 1.29 (1.09–1.52) | 1.43 (1.23–1.67) | 1.31 (1.10–1.56) |
| Sex-discrimination | | | | | | | | | |
| Not discriminated | 18735 | 1 (ref.) | 1 (ref.) | 1 (ref.) | 1 (ref.) | 1 (ref.) | 1 (ref.) | 1 (ref.) | 1 (ref.) |
| Onset of discrimination | 281 | 1.18 (0.66–2.13) | 1.45 (0.83–2.51) | 1.23 (0.67–2.26) | 1.48 (0.83–2.64) | 0.90 (0.65–1.25) | 1.30 (0.99–1.70) | 1.12 (0.85–1.48) | 1.15 (0.85–1.55) |
| Discrimination because of ethnicity | | | | | | | | | |
| Not discriminated | 18691 | 1 (ref.) | 1 (ref.) | 1 (ref.) | 1 (ref.) | 1 (ref.) | 1 (ref.) | 1 (ref.) | 1 (ref.) |
| Onset of discrimination | 41 | 1.27 (0.31–5.23) | 0.72 (0.11–4.65) | 2.93 (1.13–7.56) | 3.20 (1.21–8.48) | 0.99 (0.52–1.89) | 0.98 (0.50–1.92) | 1.33 (0.78–2.29) | 2.12 (1.43–3.13) |
| Discrimination because of education | | | | | | | | | |
| Not discriminated | 18719 | 1 (ref.) | 1 (ref.) | 1 (ref.) | 1 (ref.) | 1 (ref.) | 1 (ref.) | 1 (ref.) | 1 (ref.) |
| Onset of discrimination | 792 | 1.40 (1.02–1.90) | 1.49 (1.08–2.04) | 2.13 (1.62–2.79) | 1.44 (1.02–2.04) | 0.98 (0.83–1.16) | 1.12 (0.96–1.31) | 1.15 (0.99–1.33) | 1.18 (1.01–1.38) |
| Discrimination because of position | | | | | | | | | |
| Not discriminated | 18668 | 1 (ref.) | 1 (ref.) | 1 (ref.) | 1 (ref.) | 1 (ref.) | 1 (ref.) | 1 (ref.) | 1 (ref.) |
| Onset of discrimination | 674 | 1.15 (0.79–1.68) | 1.24 (0.84–1.82) | 2.04 (1.50–2.78) | 1.70 (1.19–2.42) | 1.19 (1.01–1.40) | 1.17 (0.99–1.38) | 1.26 (1.10–1.47) | 1.21 (1.02–1.44) |
| Discrimination because of opinion | | | | | | | | | |
| Not discriminated | 18701 | 1 (ref.) | 1 (ref.) | 1 (ref.) | 1 (ref.) | 1 (ref.) | 1 (ref.) | 1 (ref.) | 1 (ref.) |
| Onset of discrimination | 1081 | 1.38 (1.05–1.83) | 1.98 (1.54–2.54) | 2.38 (1.88–3.01) | 1.65 (1.24–2.21) | 1.25 (1.10–1.42) | 1.20 (1.05–1.36) | 1.22 (1.07–1.38) | 1.26 (1.10–1.45) |
| Additional adjustment for prior long-term sickness absence | | | | | | | | | |
| Not discriminated | 37440 | 1 (ref.) | 1 (ref.) | 1 (ref.) | 1 (ref.) | 1 (ref.) | 1 (ref.) | 1 (ref.) | 1 (ref.) |
| Onset of discrimination | 1973 | 1.08 (0.81–1.45) | 1.25 (0.93–1.68) | 2.03 (1.60–2.57) | 1.80 (1.38–2.36) | 1.13 (1.00–1.28) | 1.15 (1.02–1.31) | 1.25 (1.11–1.41) | 1.17 (1.03–1.34) |

[a] Log-binomial regression analyses adjusted for the following variables at year 0: Age, sex, chronic disease, psychological distress, anxiety, BMI, alcohol consumption, shift work, employment contract, occupational grade, work-unit size, work-unit temporary employment, work-unit gender distribution as well as data-cycle no.

In addressing the potential effect of discrimination on turnover, results pointed to a higher risk of employees working less than 50% of the time in the year following exposure to discrimination (OR = 1.34; 95% CI 1.15–1.56) indicating that employees have left work sometime during that year (S3 Table).

Sensitivity analyses with the full dataset with imputed values for missing covariates replicated the main findings (S3 Fig and S4 Table).

## Discussion

In robust models determining both between and within subject differences in LTSA associated with WD, we found that perceived WD was associated with a higher risk of LTSA due to mental, but not non-mental conditions in a large-scale cohort study of Finnish public sector employees. This excess risk was highest for concurrent LTSA during the year when WD was reported, and was particularly strong for discrimination due to ethnicity.

**Table 4. Changes in cause-specific workplace discrimination and odds for concurrent and 1-year lagged long-term sickness absence.**

| | Mental diagnosed | | | | Non-mental diagnosed | | | |
|---|---|---|---|---|---|---|---|---|
| | Concurrent sickness absence[a] | | 1-year lagged sickness absence[b] | | Concurrent sickness absence[a] | | 1-year lagged sickness absence[b] | |
| | Observations | OR [c] (95% CI) | No of observations | OR [c] (95% CI) | No of observations | OR [c] (95% CI) | N of observations | OR [c] (95% CI) |
| Cause-specific workplace discrimination | | | | | | | | |
| Age-discrimination | 1766 | 2.30 (1.19–4.43) | 1578 | 0.88 (0.49–1.58) | 7896 | 1.05 (0.78-1-40) | 7588 | 1.22 (0.95–1.65) |
| Sex-discrimination | 1694 | 1.49 (0.67–3.33) | 1522 | 0.62 (0.27–1.45) | 7566 | 0.91 (0.58–1.43) | 7318 | 1.33 (0.84–2.10) |
| Discrimination because of ethnicity | 1636 | 10.56 (1.36–82.14) | 1454 | 1.17 (0.23–5.85) | 7380 | 0.97 (0.45–2.09) | 7130 | 3.09 (1.13–8.44) |
| Discrimination because of education | 1830 | 1.91 (1.18–3.11) | 1640 | 1.09 (0.66–1.81) | 7990 | 1.00 (0.77-1-31) | 7738 | 1.04 (0.81–1.35) |
| Discrimination because of position | 1818 | 1.77 (1.13–2.76) | 1600 | 1.09 (0.66–1.81) | 7916 | 0.99 (0.76–1.29) | 7668 | 1.24 (0.95–1.61) |
| Discrimination because of opinion | 2026 | 1.95 (1.38–2.74) | 1798 | 0.67 (0.47–0.96) | 8294 | 1.00 (0.81–1.23) | 8082 | 1.12 (0.91–1.38) |
| Additional adjustment for long-term sickness absence one year prior | | | | | | | | |
| Discrimination | - | - | 6621 | 0.97 (0.79–1.19) | - | - | 26,906 | 1.11 (0.99–1.24) |
| Additional adjustment for alcohol consumption, BMI and psychological distress | | | | | | | | |
| Discrimination | 6576 | 1.62 (1.33–1.98) | 6351 | 0.98 (0.79–1.20) | 25,655 | 1.05 (0.94–1.18) | 25,767 | 1.09 (0.97–1.23) |

[a] Conditional logistic regression analyses for the following variables at year 0: Age, sex, chronic disease, psychological distress, anxiety, BMI, alcohol consumption, shift work, employment contract, occupational grade, work-unit size, work-unit temporary employment, work-unit gender distribution as well as data-cycle no.

Our results are in accordance with the theoretical hypothesis that workplace discrimination may act as an uncontrollable social stressor triggering stress reactions [9], which may manifest in mental disorders causing spells of LTSA–effects which are evident already within a year of exposure in our analyses. The adverse effect of WD on mental LTSA found in our study is in line with findings from previous cross-sectional studies on self-reported sickness absence [12,13,18–22]. Our findings are also in line with those of a longitudinal study of WD and sickness absence, which addressed age-discrimination only and showed higher rates of sickness absence after a 3-year follow-up among age-discriminated employees [17] and support the increasingly popular view that stressors often act as disease triggers [41].

We add to this literature by looking into all cause as well as specific types of discrimination and by distinguishing between LTSA due to diagnosed mental as well as non-mental conditions. This distinction seem important for understanding the relationship between WD and LTSA. It is for example interesting that we, in a subsample of the population, found that the potential effects are the strongest for discrimination due to ethnicity with higher rates of concurrent mental LTSA, and also higher rates of non-mental LTSA in the year following exposure. Our results also support a general detrimental effect of discrimination on LTSA due to mental disorders, irrespective the cause of discrimination with one exception. We did not observe associations with sex-discrimination which may be due to composition of our study population comprising primarily women.

According to a large-scale review, there is substantial support for an adverse effect of discrimination (not addressing WD per se) on self-reported mental health outcomes such as depressive symptoms, distress and low well-being, although few studies have determined these association using objective measures of diagnosed mental illness [9]. Thus, we add to current

evidence by examining timing and changes in workplace discrimination and by showing a strong short-term association between workplace discrimination and medically certified spells of LTSA with an underlying mental diagnosis. The more pronounced associations with mental than non-mental diagnoses of sickness absence support previous studies on discrimination both within [21,42] and outside the workplace setting, pointing to acute or triggering health effects, which are stronger for mental than physical health outcomes [9].

This study has several strengths. The study was based on a large cohort of Finnish public sector employees from a wide range of occupations. The prospective design, use of repeated measures of workplace discrimination and objective and universally covering measure of LTSA including the underlying diagnoses confer other strengths of the study. The use of fixed effect analyses reduced concerns of bias from person-stable characteristics, while the assessment of the effect of onset of discrimination reduced problems of selection. With our longitudinal study design, we were able to minimize reverse causality as a source of bias.

Whereas previous studies on workplace discrimination and health outcomes have primarily been based on self-reported measures of both exposure and outcome, conferring risk of common method bias, we have used a universal and objective measure of LTSA from national registers. The Finnish Social Insurance Institution keeps records on all sickness allowances paid for medically certified LTSA spells of nine or more days for the entire population, thus ensuring a valid record of LTSA independent of the reporting of workplace exposures.

Further, the majority of previous studies have examined the association with exposure at an arbitrary time-point not taking into account that some employees exposed to workplace discrimination will leave their workplace as a means of coping. A mechanism also supported by our findings. To reduce this issue of selection, we have determined the risk of LTSA in association with the onset of discrimination in a study sample free of this exposure at baseline. However, information on workplace discrimination was only available in two year intervals and we have no data on when between two waves the workplace discrimination began. Therefore, to better explicate potential trends in LTSA, we assessed the association with the full trajectory of LTSA covering the period from baseline assessment to one year after reported onset of workplace discrimination. Concurrent and 1-year lagged LTSA was assessed along with the full trajectory of LTSA in the four-year period covering each of the study-cycles, thus providing information on potential acute/triggering and prolonged effects of workplace discrimination. Further, to better accommodate common concerns of person-stable characteristics confounding the relation between workplace exposures and health outcomes, fixed-effects analyses were carried out using each employee as his or her own control.

Some limitations need to be considered when interpreting our findings. First, perceived WD was measured using a single question rather than a multi-item scale. In addition, we had only limited information on the characteristics of WD; for example, no data were available about the source (leader, organization, co-worker), frequency or perceived severity of discrimination. However, the cause of WD (age, sex, ethnicity, education, opinion, or position) was requested in a sub-sample, allowing analyses examining this factor as a potential source for heterogeneity in associations. This adds new insights as previous studies have typically collapsed multiple causes of discrimination into single measure or focused on only a single cause of discrimination, such as age, race or sex [12,17,42,43]. Further studies are warranted to elaborate how different types and sources of WD may be associated with LTSA.

Second, we focused on LTSA in the current study, which have proven a good measure of employee health as well as workplace productivity and commitment. However, workplace discrimination may in addition affect the occurrence of shorter-term absences and/or the total duration of absence. To assess the extent to which workplace discrimination affects exit from the labour market, we carried out a sensitivity analysis of the risk of working less than 50% of

the time in the year following onset of discrimination. These results support turnover as a means of coping, thus highlighting an important pit falls of the majority of current evidence on the relation between discrimination and LTSA, which includes only one baseline measure of discrimination neglecting the importance of the timing i.e. duration of exposure.

Lastly, the population comprised public sector employees, primarily female, and the results may not extrapolate directly to other settings such as the private sector in which the context may be different.

In conclusion, we found that workplace discrimination was associated with a higher risk of long-term sickness absence with an underlying mental diagnosis. This excess risk was primarily restricted to spells of sickness absences occurring during the year where workplace discrimination was reported, pointing to potential acute or triggering effects. This study highlights the importance of identifying and combating discriminatory practices at the workplace. We found that self-reported workplace discrimination predisposed employees to health problems and long-term sickness absence, thus being a health risk factor for the employee and a source of burden for employers and organizations in terms of decreased productivity. In Finland the government has introduced a new Non-Discrimination Act in 2014 [44], emphasizing the importance of prevention of discrimination at work. Future studies investigating the impact of this and similar law enforcement as an effective means of reducing long-term sickness absence are warranted.

## Supporting information

**S1 Fig. Flowchart showing the selection of eligible FPS employees for the analyses of onset of and changes in workplace discrimination and long-term sickness absence.** N; number of employees, n; number of employee-observations, LTSA; long-term sickness absence.
(PDF)

**S2 Fig. Directed acyclic graph of the assumed causal network between work-place discrimination and long-term sickness absence.**
(DOCX)

**S3 Fig. Sensitivity analysis by using imputed data on missing covariates.** Prevalence of long-term sickness absence among employees with and without onset of workplace discrimination as well as the risk ratio (RR) and 95% CI's associated with onset of workplace discrimination.
(DOCX)

**S1 Table. Measurement time-points for the analyses of onset of and changes in workplace discrimination from the Finnish Public Sector Study.**
(PDF)

**S2 Table. Supplementary analysis.** Prior long-term sickness absence and odds for onset of discrimination.
(DOCX)

**S3 Table. Supplementary analysis.** Onset of workplace discrimination and odds for turnover following exposure.
(DOCX)

**S4 Table. Sensitivity analysis by using imputed data on missing covariates.** Changes in workplace discrimination and odds for concurrent and 1-year lagged long-term sickness absence.
(DOCX)

## Author Contributions

**Conceptualization:** Alice Clark, Sari Stenholm, Jaana Pentti, Paula Salo, Theis Lange, Eszter Török, Tianwei Xu, Jesper Fabricius, Tuula Oksanen, Mika Kivimäki, Jussi Vahtera, Naja Hulvej Rod.

**Data curation:** Jaana Pentti.

**Formal analysis:** Alice Clark, Jaana Pentti, Jussi Vahtera, Naja Hulvej Rod.

**Funding acquisition:** Mika Kivimäki, Jussi Vahtera, Naja Hulvej Rod.

**Supervision:** Naja Hulvej Rod.

**Writing – original draft:** Alice Clark, Sari Stenholm, Jussi Vahtera, Naja Hulvej Rod.

**Writing – review & editing:** Alice Clark, Sari Stenholm, Jaana Pentti, Paula Salo, Theis Lange, Eszter Török, Tianwei Xu, Jesper Fabricius, Tuula Oksanen, Mika Kivimäki, Jussi Vahtera, Naja Hulvej Rod.

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
