## [Decision Letter · Decision Letter 0]

30 Apr 2021

PONE-D-21-07363

Workplace discrimination as risk factor for long-term sickness absence: Longitudinal analyses of onset and changes in workplace adversity

PLOS ONE

Dear Dr. Stenholm,

Thank you for submitting your manuscript to PLOS ONE. After careful consideration, we feel that it has merit but does not fully meet PLOS ONE’s publication criteria as it currently stands. Therefore, we invite you to submit a revised version of the manuscript that addresses the points raised during the review process.

We look forward to receiving your revised manuscript.

Kind regards,

Ali B. Mahmoud, Ph.D.

Academic Editor

PLOS ONE

Journal Requirements:

4. Please amend the manuscript submission data (via Edit Submission) to include author Tianwei Xu.

Reviewers' comments:

Reviewer's Responses to Questions

**Comments to the Author**

1. Is the manuscript technically sound, and do the data support the conclusions?

Reviewer #1: No

Reviewer #2: Yes

Reviewer #3: Yes

2. Has the statistical analysis been performed appropriately and rigorously? 

Reviewer #1: I Don't Know

Reviewer #2: Yes

Reviewer #3: No

3. Have the authors made all data underlying the findings in their manuscript fully available?

Reviewer #1: Yes

Reviewer #2: Yes

Reviewer #3: No

4. Is the manuscript presented in an intelligible fashion and written in standard English?

Reviewer #1: No

Reviewer #2: Yes

Reviewer #3: Yes

5. Review Comments to the Author

Reviewer #1: Thanks for the opportunity to review this paper. I have several queries/clarifications set out below.

Methods

Overall I found the methods and results sections very difficult to read. Many of your paragraphs explaining your methods were just not clear. The problem is compounded possibly because of your use of run on sentences - sentences which run to 5/6/7 lines - and which become increasingly incomprehensible the longer they are. You could try splitting these long sentences into shorter sentences which might improve comprehension. See for example lines 332-337.

Line 126 - you suddenly start discussing number of observations instead of number of participants - and this is displayed in some of your tables. Why have you done this - I could see no explanation provided in your manuscript?

Line 139 - has the workplace discrimination question been used/validated elsewhere? What type of discrimination is it measuring?

Paragraph commencing line 155 is not clear - perhaps you can show this in a diagram? And how do you know your participants didn't experience discrimination in year 0 - was this because the data wasn't collected?

Similarly the paragraph commencing on line 169 is unclear.

Line 172 - what was your cut off for psychological distress using the GHQ? Ditto the TAI?

Results

Line 274. These sentences are not clear - they don't seem to relate to figure 1 - which has more than you are describing in this paragraph - and appears to be set out differently to your depiction?

Reviewer #2: Thank you for the opportunity to review the work entitled “Workplace discrimination as risk factor for long-term sickness absence: Longitudinal analyses of onset and changes in workplace adversity” This paper focuses on a relevant and important topic. In particular, I appreciate the aim to examine the onset of workplace discrimination (WD) and changes of WD on long-term sickness absence (LTSA). I think the methods are convincing. However, the paper has some areas for improvement and clarification. Please see my comments below.

1.The Introduction is well written. The authors have emphasized the lack of longitudinal studies on this topic and the methodological issues of previous studies, however, I believe that the authors need to explain more the rationale behind the study in the introduction section. In the introduction, it’s unclear how workplace discrimination is different from other working environment factors, such as job stress, job demand, or job support. Why workplace discrimination is a practical factor that the authors want to look at? In the discussion, the authors mentioned that workplace discrimination is an uncontrollable social stressor triggering stress reactions, but this ‘social stressor’ theory was not mentioned in the Introduction. Also, I think the introduction can benefit from explaining why focusing on LTSA? What is the government policy for LTSA in Finland, and whether the policy for LTSA is the same for those working in the public sector and those who work in other sectors?

2. Table 1 needs some clarification. For example, the authors said the study population is N=32519, but this number is not shown in Table 1. Also, as described “Of the 29597 participants who were not discriminated at baseline (39,413 employee-observations), 5.0% reported WD at the second time-point.” But this information is not clearly presented in Table 1.

Reviewer #3: Thank you for the opportunity to review this interesting manuscript. Overall the details were presented coherently to approach the aim of the proposed study. I have a few minor comments for the authors to consider:

• Abstract:

o Please clarify if the RR or OR (line 48, line 51) were adjusted or crude ones. It would be good to clarify so by adding ‘adjusted’ or ‘crude’ to the RR or OR.

• Introduction:

o It would be good to clarify the setting of the study, e.g. Finnish evidence around this research topic is lacking, to link the research gap to the research aim. This clarification could be added to the beginning of the last paragraph (line 90), to set the scope of this study so that the readers are not expected worldwide data.

• Methods:

o Line 132: The authors mentioned that “observations with missing information on time-varying covariates (n=1125) were excluded”. Please justify a reason to not perform analyses using the “missing by random” (or other) technique.

o Line 175: The authors mentioned that “directed acyclic graphs” method was used to identify potential confounders. Please include another supplementary graph of the DAG.

o Sensitivity analyses (line 221 onwards): This may be renamed as ‘subgroup analyses’ or ‘supplementary analyses’. In saying that, it would be good to have a sensitivity analyses done on the full dataset and on the missing data dataset (or simulated for the missing variables) to assure the readers if there is any difference in the finding.

o Please clarify if the level of significance was set at 5% (or 1% or lower due to the multiple comparisons).

• Results:

o It would be good to also specify the analyses done (e.g. logistic regression) in the footnotes of the tables (e.g., added to footnote c of Table 2, footnote a of Table 3, etc)

o Sensitivity analyses (line 343 onwards): please update as per suggestions under the Methods section.

o Figure 1: Please add analyses done as a footnote too.

• Discussion

o The limitations of the study may be grouped into a long paragraph, after the paragraph around the strengths of the study.

o The last sentence in the Discussion section (line 445-446: “With our longitudinal study design, we were able to minimize reverse causality as a source of bias.”) could be placed elsewhere (e.g. in line 397) so that the ‘strengths’ of the study go together in a paragraph.

o It would be good to add a sentence around the implication of this research finding (or a few sentences around suggestions for future studies) as the concluding remark.

6. PLOS authors have the option to publish the peer review history of their article (what does this mean?). If published, this will include your full peer review and any attached files.

Reviewer #1: No

Reviewer #2: No

Reviewer #3: No

---

## [Author Response · Author response to Decision Letter 0]

16 Jun 2021

Response to Reviewers’ comments:

Reviewer 1

COMMENT 1: 

Thanks for the opportunity to review this paper. I have several queries/clarifications set out below.

RESPONSE: We thank the Reviewer for useful comments. Please see our point-by-point responses below. 

COMMENT 2: 

Overall I found the methods and results sections very difficult to read. Many of your paragraphs explaining your methods were just not clear. The problem is compounded possibly because of your use of run on sentences - sentences which run to 5/6/7 lines - and which become increasingly incomprehensible the longer they are. You could try splitting these long sentences into shorter sentences which might improve comprehension. See for example lines 332-337.

RESPONSE: Thank you for this suggestion. We have now revised the Methods and Results section by splitting long sentences into shorter ones and by clarifying the sentences. For example, the sentence highlighted by the Reviewer is revised as follows: 

“In terms of LTSA due to non-mental conditions, risk of LTSA was consistently slightly higher among employees with onset of WD due to age, position and opinion compared to those without onset of WD (Table 3). Results pointed to a higher prevalence of LTSA following onset of WD due to ethnicity, with a twofold risk of LTSA at year 3 (RR=2.12; 95% CI:1.43-3.13).” (page 16, lines 338-342)

(WD = workplace discrimination; LTSA = long-term sickness absence)

COMMENT 3: 

Line 126 - you suddenly start discussing number of observations instead of number of participants - and this is displayed in some of your tables. Why have you done this - I could see no explanation provided in your manuscript?

RESPONSE: We agree that this was not clearly described in the original version. There were four data collection waves in the FPS study and participating in two or more waves was required to be included in the analysis. Since some participants had responded to all four waves, they contributed to two data cycles (one cycle includes two waves) and thus two observations as illustrated in Supplemental Table 1. Therefore, the inclusion criteria were fulfilled by 32,519 employees. Some of these participants contributed with several data cycles, totalling of 44,704 observations. This is now illustrated in the Supplemental Figure 1.

We have modified the description of the study population in the Methods section as follows:

“Since some participants had participated in all four study waves, they contributed observations to two separate two-wave data cycles (S1 Table). Given this, the 32,519 participants contributed with a total of 44,704 observations.” (page 5, lines 126-128)

COMMENT 4: 

Line 139 - has the workplace discrimination question been used/validated elsewhere? What type of discrimination is it measuring?

RESPONSE: Thank you for this comment. The question about workplace discrimination has been included in the Finnish Public Sector survey since 1997. It originates from the Statistics Finland’s workplace climate survey, which was conducted for the first time in 1991 (Statistics Finland 1991). The question measures participants’ general experiences on discrimination at their workplace. Since 2008 , information regarding potential causes of workplace discrimination has also been inquired (age, sex, ethnicity, education, opinion, and position). However, detailed characteristics of workplace discrimination is unknown, such as the source (leader, organization, co-worker), frequency or perceived severity of discrimination.

We have added a reference to the workplace discrimination measure in the Methods section (page 6) and commented it in the Discussion as follows:

“First, perceived WD was measured using a single question rather than a multi-item scale. In addition, we had only limited information on the characteristics of WD; for example, no data were available from the source (leader, organization, co-worker), frequency, or perceived severity of discrimination. However, the cause of WD (age, sex, ethnicity, education, opinion, or position) was requested in a sub-sample, allowing for analyses examining this factor as a potential source for the heterogeneity in associations. This adds new insights as previous studies have typically collapsed multiple causes of discrimination into a single measure or focused on only a single cause of discrimination, such as sex or race (12, 17, 42, 43). Further studies are warranted to elaborate on how different types and sources of WD may be associated with LTSA.” (page 23, lines 438-448)

Reference: 

Statistics Finland scale on workplace climate. Lehto A-M. Quality of working life and equity. Helsinki, Finland: Statistics Finland, 1991.

COMMENT 5: 

Paragraph commencing line 155 is not clear - perhaps you can show this in a diagram? And how do you know your participants didn't experience discrimination in year 0 - was this because the data wasn't collected? Similarly the paragraph commencing on line 169 is unclear.

RESPONSE: Thank you for these comments, we are happy to clarify. In fact, we show the study design and respective measurement points in the S1 Table. 

Regarding the analyses of the onset of WD, the inclusion criteria was that the participant at baseline (year 0, i.e. Tx) reported not to be exposed to discrimination. According to the pseudo trial design, some participants reported onset of discrimination and some participants remained free of discrimination at year 2 (i.e. Tx+1). The outcome, prevalence of sickness absence, was assessed annually from year 0 to year 3 (i.e. Tx+2).

Regarding the analyses of change in WD, two different LTSA outcomes were defined. First, a concurrent LTSA was assessed within the year pertaining to the reporting of exposure, that is WD. Second, a 1-year lagged LTSA was assessed as spells initiated during the year following the assessment of exposure.

We now provide a more detail description for both analytical approaches in the Methods section:

“In the analyses with onset of WD as the exposure, the annual risk of LTSA (one or more episodes) for the groups with and without onset of WD was assessed. The risk of LTSA was calculated from the year pertaining to the first reporting of WD i.e. year 0 (where no employees were exposed to discrimination) to year 3 (one year after the second study wave when WD was assessed). The study design is illustrated in the S1 Table.

In analyses with change in WD as the exposure, two different LTSA outcomes were defined: Concurrent LTSA was defined as sickness absence spells within the year pertaining to the reporting of exposure, and 1-year lagged LTSA was defined as sickness absence spells initiated during the year that followed the assessment of exposure. The study design is illustrated in the S1 Table.” (page 7, lines 161-170)

COMMENT 6: 

Line 172 - what was your cut off for psychological distress using the GHQ? Ditto the TAI?

RESPONSE: As suggested by Goldberg 1972, the cut off for psychological distress scale was four or more symptoms. For the Trait Anxiety Inventory, we used a continuous scale based on the average response score from the six-item scale. 

This is now clarified in the Methods section as follows:

“Psychological distress was assessed using the 12-item General Health Questionnaire and the score was dichotomized (<4 vs ≥4 symptoms) (38). The six-item Trait Anxiety Inventory (39) was used to assess liability to anxiety and was analyzed on a continuous scale based on the average of responses 1-4.“ (page 8, lines 184-187)

References: 

Goldberg DP. The detection of psychiatric illness by questionnaire; a technique for the identification and assessment of non-psychotic psychiatric illness. London: Oxford University Press 1972.

Marteau TM, Bekker H. The development of a six‐item short‐form of the state scale of the Spielberger State—Trait Anxiety Inventory (STAI). Br J Clin Psychol. 1992;31(3):301–6.

COMMENT 7: 

Results: Line 274. These sentences are not clear - they don't seem to relate to figure 1 - which has more than you are describing in this paragraph - and appears to be set out differently to your depiction?

RESPONSE: We apologise that the description was unclear. We have now revised the text in the Results section to improve clarity:

“The prevalence of LTSA differed markedly between the groups during the 4-year follow-up (Figure 1). Among employees not experiencing WD, the prevalence of LTSA due to a mental disorders was almost constant over the 4-year period. Meanwhile, in employees with onset of WD there was an increase in LTSA due to a mental health disorders already at year 1. This difference in LTSA levels between the two exposure groups reached its peak at year 2, the time point when the onset of WD was reported (RR=2.13; 95% CI 1.80-2.52).” (page 13, lines 293-298)

 

Reviewer 2

COMMENT 1: 

Thank you for the opportunity to review the work entitled “Workplace discrimination as risk factor for long-term sickness absence: Longitudinal analyses of onset and changes in workplace adversity” This paper focuses on a relevant and important topic. In particular, I appreciate the aim to examine the onset of workplace discrimination (WD) and changes of WD on long-term sickness absence (LTSA). I think the methods are convincing. However, the paper has some areas for improvement and clarification. Please see my comments below.

RESPONSE: Thank you for the encouraging comments and useful suggestions. Please see our point-by-point responses below.

COMMENT 2: 

The Introduction is well written. The authors have emphasized the lack of longitudinal studies on this topic and the methodological issues of previous studies, however, I believe that the authors need to explain more the rationale behind the study in the introduction section. In the introduction, it’s unclear how workplace discrimination is different from other working environment factors, such as job stress, job demand, or job support. Why workplace discrimination is a practical factor that the authors want to look at? In the discussion, the authors mentioned that workplace discrimination is an uncontrollable social stressor triggering stress reactions, but this ‘social stressor’ theory was not mentioned in the Introduction. Also, I think the introduction can benefit from explaining why focusing on LTSA? What is the government policy for LTSA in Finland, and whether the policy for LTSA is the same for those working in the public sector and those who work in other sectors?

RESPONSE: Thank for these important comments. As suggested by the Reviewer, we have provided a more comprehensive description about the conceptual differences between workplace discrimination and other psychosocial work characteristics in the Introduction as follows:

“A large body of research has linked various adverse psychosocial work characteristics, including job strain, effort-reward imbalance, job insecurity and long working hours, to increased risk of physical and mental chronic conditions (1-5). These factors characterize unhealthy features of work tasks and organization of work. In addition, negative interpersonal relations at work have been hypothesized to predispose employees to ill health.” (page 3, lines 56-60)

We have also emphasized the importance to investigate workplace discrimination, as it is a common social stressor at work:

“Workplace discrimination is a specific social stressors, being typically both unpredictable and uncontrollable. Among the general working population in Finland, 11% of women and 6% of men reported experiencing discrimination at their workplace (6).” (page 3, lines 61-63)

We have also added a justification for LTSA as an outcome as follows:

“Further, prior studies point to a potential link between workplace discrimination and the risk of sickness absence (12, 13, 17-22). Sickness absence, especially when long-term, is considered as a measure of employee health and well-being as well as a correlate of workplace productivity (23-26).’’ (page 3, lines 67-70)

We have also specified the policies related to LTSA in the Methods as follows:

“Using employees’ unique personal identification numbers, we linked participants to the records of medically-certified LTSA from the Finnish Social Insurance Institution register, including dates and diagnoses for LTSA spells lasting 10 days or more. This register covers the entire working population including public and private sector, but does not include sickness absence due to maternity leave and caring for a sick child.” (page 7, lines 153-157)

COMMENT 3: 

Table 1 needs some clarification. For example, the authors said the study population is N=32519, but this number is not shown in Table 1. Also, as described “Of the 29597 participants who were not discriminated at baseline (39,413 employee-observations), 5.0% reported WD at the second time-point.” But this information is not clearly presented in Table 1.

RESPONSE: Thank you for these important comments. For the sake of clarity, we have now presented baseline characteristics for the study sample included in the “Onset of workplace discrimination” analysis. There were a total of 39,413 employee-observations obtained from 29,597 participants. We have revised the text as follows:

“Among the 39,413 employee-observations from the group of 29,597 participants not exposed to WD at the baseline, 1,973 (5%) employee-observations among 1,920 participants indicated WD at the second time-point.” (page 12, lines 257-260)

 

Reviewer 3

COMMENT 1: Thank you for the opportunity to review this interesting manuscript. Overall the details were presented coherently to approach the aim of the proposed study. I have a few minor comments for the authors to consider:

RESPONSE: Thank you for the positive comments and good suggestions. Please see our point-by-point responses below.

COMMENT 2: Abstract: Please clarify if the RR or OR (line 48, line 51) were adjusted or crude ones. It would be good to clarify so by adding ‘adjusted’ or ‘crude’ to the RR or OR.

RESPONSE: As suggested by the Reviewer, we have clarified in the abstract that RRs and ORs are based on adjusted models.

COMMENT 3: Introduction: It would be good to clarify the setting of the study, e.g. Finnish evidence around this research topic is lacking, to link the research gap to the research aim. This clarification could be added to the beginning of the last paragraph (line 90), to set the scope of this study so that the readers are not expected worldwide data.

RESPONSE: Thank you for this suggestion. We have supplemented the last paragraph in the Introduction to address also context of the study as follows:

“To address these two methodological shortcomings, we combined longitudinal survey with register data from national registries from Finland and applied robust methodological approaches in assessing the associations of onset of WD and changes in WD with risk of subsequent long-term sickness absence (LTSA).” (page 4, lines 95-98)

COMMENT 4: Methods: Line 132: The authors mentioned that “observations with missing information on time-varying covariates (n=1125) were excluded”. Please justify a reason to not perform analyses using the “missing by random” (or other) technique.

RESPONSE: Thank you for this comment. Indeed, we excluded observations with missing information on covariates. However, 1125 represents only 1.3% of the entire number of observations (n=89,408) included in the “Change in workplace discrimination” analysis. This is also shown in the flowchart S1 Fig. Due to the very small proportion of observations, excluding them did not influence the results of this study. 

To illustrate the role of missing observations in the main findings, we imputed values for missing covariates and repeated the main analysis. These results are now shown in the S2 Figure and S4 Table indicating that the main results were replicated and only minor differences in the second decimal of the RRs or ORs and their 95% CIs were observed.

COMMENT 5: Line 175: The authors mentioned that “directed acyclic graphs” method was used to identify potential confounders. Please include another supplementary graph of the DAG.

RESPONSE: We have now included the DAG in the supplementary material (S2 Figure).

COMMENT 6: Sensitivity analyses (line 221 onwards): This may be renamed as ‘subgroup analyses’ or ‘supplementary analyses’. In saying that, it would be good to have a sensitivity analyses done on the full dataset and on the missing data dataset (or simulated for the missing variables) to assure the readers if there is any difference in the finding.

RESPONSE: As suggested by the Reviewer, we have replaced term “Sensitivity analyses” by “Supplementary analyses” in the Statistical Analysis and Results sections. In addition, as suggested we have conducted sensitivity analysis with the full dataset by imputing values for missing covariates. This is now described in the Statistical Analysis section as follows:

“Finally, due to missing data on covariates, 6% of the observations for the onset analysis and 1% of the observations for the change analysis were excluded from these analyses. Sensitivity analysis for the main analyses were conducted by using mean imputation method in the full dataset in which missing covariates were replaced with their overall estimated mean.” (page 11, lines 246-249)

The results are shown in the S2 Figure and S4 Table indicating that the main results were replicated and only minor differences in the second decimal of the RRs or ORs and their 95% CIs were observed. This is also mentioned in the Results section as follows:

“Sensitivity analyses with the full dataset with imputed values for missing covariates replicated the main findings (S2 Figure and S4 Table).” (page 20, lines 368-369)

COMMENT 7: Please clarify if the level of significance was set at 5% (or 1% or lower due to the multiple comparisons).

RESPONSE: We have clarified this in the Statistical Analysis section as follows:

“The level of statistical significance was set at 5%.” (page 11, line 251)

COMMENT 8: Results: It would be good to also specify the analyses done (e.g. logistic regression) in the footnotes of the tables (e.g., added to footnote c of Table 2, footnote a of Table 3, etc).

RESPONSE: As suggested by the Reviewer, we have specified the analytical methods in the footnotes of the tables (Table 2, Table 3, Table 4, S2 Table, S3 Table and S4 Table).

COMMENT 9: Sensitivity analyses (line 343 onwards): please update as per suggestions under the Methods section.

RESPONSE: As suggested by the Reviewer, we have replaced term “Sensitivity analyses” by “Supplementary analyses” in the Statistical Analysis and Results sections.

COMMENT 10: Figure 1: Please add analyses done as a footnote too.

RESPONSE: As suggested by the Reviewer, we have specified the analytical methods in the Figure 1 footnote.

COMMENT 11: Discussion: The limitations of the study may be grouped into a long paragraph, after the paragraph around the strengths of the study.

RESPONSE: Thank you for this suggestion. We have grouped the limitations of the study into longer section towards the end of discussion section.

COMMENT 12: The last sentence in the Discussion section (line 445-446: “With our longitudinal study design, we were able to minimize reverse causality as a source of bias.”) could be placed elsewhere (e.g. in line 397) so that the ‘strengths’ of the study go together in a paragraph.

RESPONSE: As suggested, we have moved the sentence earlier in the Discussion (page 22, lines 413-414).

COMMENT 13: It would be good to add a sentence around the implication of this research finding (or a few sentences around suggestions for future studies) as the concluding remark.

RESPONSE: As suggested by the Reviewer, we have elaborated on the implications of the study by the end of the manuscript as follows:

“This study highlights the importance of identifying and combating discriminatory practices at the workplace. We found that self-reported workplace discrimination predisposed employees to health problems and long-term sickness absence, thus being a health risk factor for the employee and a source of burden for employers and organizations in terms of lowered productivity. In Finland the government has introduced a new Non-Discrimination Act in 2014 (44), emphasizing the importance of prevention of discrimination at work. Future studies investigating the impact of this and similar law enforcement as an effective means of reducing long-term sickness absence are warranted.” (page 24, lines 464-472)

Journal Requirements

POINT 1: Please ensure that your manuscript meets PLOS ONE's style requirements, including those for file naming. 

RESPONSE: We have modified the manuscript and it now meets the journal’s style requirements.

POINT 2: Please provide additional details regarding participant consent. In the ethics statement in the Methods and online submission information, please ensure that you have specified what type you obtained (for instance, written or verbal, and if verbal, how it was documented and witnessed). If your study included minors, state whether you obtained consent from parents or guardians. If the need for consent was waived by the ethics committee, please include this information.

RESPONSE: In the questionnaire participants were provided information about the study and response to the questionnaire was considered as a consent to participate in the study.

We have specified the participant consent in the Methods section as follows:

“Participants were provided information about the study and response to the survey questionnaire was considered as a consent to participate.” (page 5, lines 115-116)

POINT 3:. We note that you have indicated that data from this study are available upon request. PLOS only allows data to be available upon request if there are legal or ethical restrictions on sharing data publicly. For information on unacceptable data access restrictions, please see http://journals.plos.org/plosone/s/data-availability#loc-unacceptable-data-access-restrictions.

RESPONSE: The outcome of this study, long-term sickness absence, is obtained from the Finnish Social Insurance Institution’s register and linked to survey data by using the unique personal identification number. According to the Finnish law, we are not allowed to share identified sensitive health data to other researchers. We have also explained this in the Cover letter.

POINT 4: Please amend the manuscript submission data (via Edit Submission) to include author Tianwei Xu.

RESPONSE: Thank you for noticing this mistake. We have now included Tianwei Xu in the submission system.

---

## [Decision Letter · Decision Letter 1]

22 Jul 2021

Workplace discrimination as risk factor for long-term sickness absence: Longitudinal analyses of onset and changes in workplace adversity

PONE-D-21-07363R1

Dear Dr. Stenholm,

I'm pleased to inform you that your manuscript has been judged scientifically suitable for publication and will be formally accepted for publication once it meets all outstanding technical requirements.

Kind regards,

Ali B. Mahmoud, Ph.D.

Academic Editor

PLOS ONE

Additional Editor Comments (optional):

Reviewers' comments:

Reviewer's Responses to Questions

**Comments to the Author**

1. If the authors have adequately addressed your comments raised in a previous round of review and you feel that this manuscript is now acceptable for publication, you may indicate that here to bypass the “Comments to the Author” section, enter your conflict of interest statement in the “Confidential to Editor” section, and submit your "Accept" recommendation.

Reviewer #1: All comments have been addressed

Reviewer #2: All comments have been addressed

Reviewer #3: All comments have been addressed

2. Is the manuscript technically sound, and do the data support the conclusions?

Reviewer #1: Yes

Reviewer #2: Yes

Reviewer #3: Yes

3. Has the statistical analysis been performed appropriately and rigorously? 

Reviewer #1: Yes

Reviewer #2: Yes

Reviewer #3: Yes

4. Have the authors made all data underlying the findings in their manuscript fully available?

Reviewer #1: Yes

Reviewer #2: Yes

Reviewer #3: Yes

5. Is the manuscript presented in an intelligible fashion and written in standard English?

Reviewer #1: Yes

Reviewer #2: Yes

Reviewer #3: Yes

6. Review Comments to the Author

Reviewer #1: all my earlier comments were addressed appropriately..........................................................

Reviewer #2: The manuscript has been thoroughly revised and my concerns and suggestions have been addressed. In particular, the introduction and method sections are now much clearer.

Reviewer #3: Thank you for addressing my previous comments and improving the quality of this revised manuscript.

7. PLOS authors have the option to publish the peer review history of their article (what does this mean?). If published, this will include your full peer review and any attached files.

Reviewer #1: No

Reviewer #2: No

Reviewer #3: No

---

## [Editor Report · Acceptance letter]

27 Jul 2021

PONE-D-21-07363R1 

Workplace discrimination as risk factor for long-term sickness absence: Longitudinal analyses of onset and changes in workplace adversity 

Dear Dr. Stenholm:

I'm pleased to inform you that your manuscript has been deemed suitable for publication in PLOS ONE. Congratulations! Your manuscript is now with our production department. 

Kind regards, 

on behalf of

Dr. Ali B. Mahmoud 

Academic Editor

PLOS ONE